# PI3K/AKT/mTOR-Targeted Therapy for Breast Cancer

**DOI:** 10.3390/cells11162508

**Published:** 2022-08-12

**Authors:** Kunrui Zhu, Yanqi Wu, Ping He, Yu Fan, Xiaorong Zhong, Hong Zheng, Ting Luo

**Affiliations:** 1Breast Disease Center, Cancer Center, West China Hospital, Sichuan University, Chengdu 610000, China; 2Multi-Omics Laboratory of Breast Diseases, State Key Laboratory of Biotherapy, National Collaborative, Innovation Center for Biotherapy, West China Hospital, Sichuan University, Chengdu 610000, China

**Keywords:** AKT, biomarker, breast cancer, cancer therapy, mTOR, PI3K

## Abstract

Phosphatidylinositol 3-kinase (PI3K), protein kinase B (PKB/AKT) and mechanistic target of rapamycin (mTOR) (PAM) pathways play important roles in breast tumorigenesis and confer worse prognosis in breast cancer patients. The inhibitors targeting three key nodes of these pathways, PI3K, AKT and mTOR, are continuously developed. For breast cancer patients to truly benefit from PAM pathway inhibitors, it is necessary to clarify the frequency and mechanism of abnormal alterations in the PAM pathway in different breast cancer subtypes, and further explore reliable biomarkers to identify the appropriate population for precision therapy. Some PI3K and mTOR inhibitors have been approved by regulatory authorities for the treatment of specific breast cancer patient populations, and many new-generation PI3K/mTOR inhibitors and AKT isoform inhibitors have also been shown to have good prospects for cancer therapy. This review summarizes the changes in the PAM signaling pathway in different subtypes of breast cancer, and the latest research progress about the biomarkers and clinical application of PAM-targeted inhibitors.

## 1. Introduction

Phosphatidylinositol 3-kinases (PI3Ks) are lipid kinases that can fall into three classes (I, II and III) in mammals [1]. It has been extensively studied for its important functions in physiology and diseases. In particular, class I PI3K is a well-studied subtype and has been confirmed to be associated with the occurrence and development of cancer [2]. Class I PI3Ks consist of a catalytic subunit p110 (p110α, p110β, p110γ or p110δ) encoded by PIK3CA, PIK3CB and PIK3CD, respectively, and a regulatory subunit p85 (p85α, p85β and p85γ) encoded by PIK3R1, PIK3R2 and PIK3R3, respectively (Figure 1). The class I PI3K, protein kinase B (PKB/AKT) and mechanistic target of rapamycin (mTOR) pathway (PAM pathway) display abnormal activation frequently in human cancer and plays a vital part in cell survival, proliferation, motility and metabolism (Figure 2) [3,4,5].

Different from class I PI3K(PIK3C1), class II/III PI3K (PIK3C2/3) mainly phosphorylates phosphatidylinositol to phosphatidylinositol 3-phosphate (PI3P), thereby regulating autophagy and vesicular trafficking [9,10]. Class II PI3K is also involved in angiogenesis and cell migration. In breast cancer, low expression of PIK3C2 seems to increase sensitivity to chemotherapeutic drugs [11]. However, PIK3C2 is less sensitive to classical PI3K inhibitors. It warrants further studies to develop selective PIK3C2 inhibitors [12,13]. In addition, PIK3C3 has been confirmed to regulate tumor cell proliferation by inducing autophagy and regulating iron metabolism [14]. Preclinical studies have shown that targeting PIK3C3 has anti-tumor activity in breast, colorectal and prostate cancer [15,16]. This review focuses on the well-studied class I PI3K and the related PAM pathway in breast cancer.

Abnormal enhancement of the PI3K/AKT/mTOR pathway often promotes excessive cell multiplication and resistance to apoptosis, and participates in the development and progression of various tumors [17]. Disturbance of the PAM pathway is particularly usual in breast cancer, with approximately 70% of breast cancer patients having alterations in this pathway [18,19]. A huge number of experiments in vitro and in vivo have shown that inhibiting key components of the PAM pathway can inhibit cancer cell proliferation and survival, and affect tumor microenvironment, angiogenesis, cancer metastasis and metabolism, thereby exerting anti-tumor effects and overcoming endocrine therapy resistance [20,21,22,23,24]. Many of small-molecule inhibitors targeting the PAM pathway have been tested in preclinical and clinical studies. Thus far, only a few PI3K and mTOR inhibitors have been approved for the treatment of breast cancer [25].

## 2. Changes of the PAM Pathway in Different Breast Cancer Subtypes

The mechanism of abnormal activation of the PAM pathways in breast cancer includes variations in key molecules, such as amplification or overexpression of RTKs (e.g., HER2/ERBB2) and KRAS (kirsten rat sarcoma viral oncogene) mutation [26]. PIK3CA, PIK3CB and PIK3R1 mutations are frequently detected in breast cancer [22,27,28,29]. There are common p110α (PIK3CA) variants in the acidic cluster of the helical domain (E542, E545, and Q546) and the histidine residue (H1047) in the kinase domain (Figure 1). In addition, AKT1 mutations and inactivation of tumor suppressor gene PTEN, TSC1/2 (Tuberous Sclerosis Complex 1/2) or INPP4B [18,30,31,32] are involved in the abnormal activation of the PAM pathways. Some of these changes can be prognostic factors or biomarkers for targeted therapy (Table 1).

The frequency of changes in the above-mentioned genes may vary among different subtypes of breast cancer (Table 1) [47,48,49,50,51,52,53]. Breast cancer can be roughly divided into four subtypes: luminal A (60–70%), luminal B (10–20%), HER2-enriched (13–15%) and triple-negative (10–15%) [52]. In estrogen receptor (ER)+ and HER2+ breast cancer, the most common mechanism of abnormal activation of the PAM pathway is PIK3CA mutation, accounting for 47% of cases of the ER+/HER2− (luminal A) subtype, 33% of the ER+/HER2+ (luminal B) subtype, 23%–39% of the ER-/HER2+ subtype and 8–25% of the triple-negative breast cancer (TNBC) subtype [33,34,36]. In advanced and metastatic breast cancer, PI3KCA mutations may lead to chemotherapy resistance and a poor prognosis. For HER2 positive breast cancer, PIK3CA mutations are associated with worse prognosis [37,54]. Among patients receiving neoadjuvant regimens containing docetaxel, carboplatin, trastuzumab and lapatinib, those with tumors with PIK3CA/ERBB family mutations seem to develop pathologic complete response (pCR) more than those with wild-type tumors [55]. For TNBC, the most common abnormal mechanism of PAM is PTEN inactivation or downregulation, accounting for 67% of cases [56,57]. In addition, mTOR hyperphosphorylation is associated with poor outcomes of patients with stage I/II TNBC [46]. Metaplastic breast cancer is a type of TNBC. Several studies found strong enrichment in mutations of PIK3CA/PIK3R1, P53 and PTEN, and aberrations of RAS-MAPK pathways in metaplastic breast cancer [38]. In addition, PDK1 amplification is present in 20–38% of all breast cancer subtypes and is involved in aberrant activation of the PAM pathway [35,45]. AKT1 activating mutations (E17K) are also observed in 7% of ER+ metastatic breast cancer patients [42]. Therefore, it is necessary to select corresponding biomarkers according to different breast cancer subtypes to guide targeted therapy and evaluate prognosis.

Inhibiting the key nodes in PAM pathway can exert anti-tumor effects [43,44]. In HR+ breast cancer, the activation of PI3K pathway by PIK3CA mutation promotes ligand-independent ER activation, which is one of the important mechanisms of endocrine therapy resistance [53,58,59,60]. Inhibition of PI3K can delay or reverse endocrine therapy resistance and improve patient prognosis. In addition, AKT activation may confer resistance to anticancer agents such as the ER antagonists tamoxifen and fulvestrant. Combination of AKT inhibitors with tamoxifen and fulvestrant may improve their effectiveness [61,62]. Pan-mTORC1/2 inhibitors can also reverse endocrine resistance, chemoresistance and radiation resistance [63]. For patients with HER2+ breast cancer, inhibition of PI3K or mTOR can become a new therapeutic regimen after secondary resistance to anti-HER2 therapy [64,65,66]. For TNBC, PAM inhibitors may have clinical benefits in PTEN-deficient population [67,68]. Owing to the complexity and interactions among the diverse components in PAM pathway, inhibition of a single target may result in compensatory changes in other targets. After inhibition of PI3K, the mTOR pathway can be abnormally activated to counteract this effect [69,70,71]. The mTOR inhibitor may promote the expression of insulin receptor substrates, which may upregulate the AKT pathway [72].

Multiple changes in PAM pathways may co-exist in breast cancer. For example, the co-existence of PIK3CA mutation, PTEN deletion and HER2 amplification is detected in breast cancer [2,73]. Hence, inhibition of a single target may not achieve anti-tumor effects in these circumstances. This not only suggests the molecular mechanism of drug resistance in breast cancer, but also the feasibility of combined therapy. Yang et al. have shown that temsirolimus (mTORC1 inhibitor) in combination with dactolisib (dual PI3K-mTOR inhibitor) or ZSTK474 (pan-PI3K inhibitor) can collectively inhibit cancer cell growth and overcome cellular resistance to temsirolimus [74]. Tang et al. also proved that using different PAM inhibitors at the same time can reach better anti-cancer effects [75].

## 3. Preclinical and Clinical Development of PAM Pathway Inhibitors in Different Subtypes of Breast Cancer

### 3.1. PAM Inhibitors for Treating ER+/HER2− Breast Cancer

About 70% of breast cancer is ER+ and HER2− [37]. Endocrine therapy is the standard regimen for these patients [25]. Abnormal activation of the PAM pathway is one of the important reasons of endocrine resistance [59], which can be overcome or reversed by targeting pathways’ components which activated during acquired drug resistance [76,77]. Therefore, PAM pathway inhibitors have been extensively studied in this population (Table 2). PAM pathway inhibitors are classified into PI3K, AKT, mTOR, and dual PI3K-mTOR inhibitors. Alpelisib (PI3K inhibitor) and everolimus (mTOR inhibitor) have been approved by the U.S. Food and Drug Administration (FDA) for the clinical treatment of breast cancer [25]. Many AKT and mTOR inhibitors have initially shown preclinical activity or are currently undergoing clinical trials.

#### 3.1.1. PI3K Inhibitors for Treating ER+/HER2− Breast Cancer

PI3K inhibitors can fall into three categories. The first category includes pan-PI3K inhibitors, which non-selectively act on the ATP-binding pockets of all class I PI3K isoforms (p110α, p110β, p110γ, and p110δ) [89]. Both buparisib and pictilisib are pan-PI3K inhibitors. Phase III clinical trials show that, as the second-line treatment of ER+ and HER2− advanced or metastatic breast cancer patients, buparisib combined with fulvestrant can significantly improve PFS, while the most frequent grade 3–4 adverse events are elevated alanine aminotransferase, hyperglycemia, hypertension and fatigue [78,79]. These severe adverse reactions may result in discontinuation of the drug. Due to the high blood–brain barrier-penetrating properties of buparlisib, depression and anxiety are also common psychiatric side-effects. Pictilisib, another pan-PI3K inhibitor, did not result in a significant survival benefit [80,81]. However, these studies consider that combining PI3K inhibition with endocrine therapy is reasonable in patients with ER +/HER2− breast cancer [36,90] (Table 2).

The second class of specific PI3K inhibitors are represented by the drugs alpelisib (BYL719) and taselisib (GDC-0032). The ATP binding sites of type I PI3K are highly homologous. The different residues next to the ATP binding sites can be divided into the adjacent hinge region (four residues) and the variable region located at the p-loop. Non-conserved residues in these two regions are key to the subtype selectivity of pan-PI3K inhibitors. Alpelisib can form a dihydrogen bond with Q859 in the hinge region of PI3Kα, while other subunits at this site are too short to form the same structure, so it can selectively inhibit p110α [91]. So far, alpelisib is the only PI3K inhibitor approved by the FDA for breast cancer treatment. In the phase III SOLAR-1 trial, ER+/HER2− advanced breast cancer patients with relapsed or progressed disease after endocrine therapy received alpelisib and fulvestrant treatment. This trial showed that alpelisib plus fulvestrant demonstrated better overall efficacy compared with placebo (26.6% vs. 12.8%). Furthermore, in patients with PIK3CA mutation, combination of alpelisib and fulvestrant greatly prolonged the mPFS (*p* = 0.001, Table 2). In contrast, fulvestrant plus alpelisib had no PFS benefit (7.4 months vs. 5.6 months, HR: 0.85; 95% CI: 0.58–1.25) in the PIK3CA wild-type group [81,82]. The positive results of SOLAR-1 trial led to the approvement of alpelisib plus fulvestrant for treating advanced or metastatic breast cancer with ER expression and PIK3CA mutation [25]. The recently updated American Society of Clinical Oncology (ASCO) guideline recommended that patients with locally recurrent unresectable or metastatic hormone receptor positive and HER2-negative breast cancer should be subject to testing of PIK3CA mutations to determine their eligibility for treatment with the alpelisib plus fulvestrant [92]. The BYLieve phase II trial assessment of the effectiveness and security of alpelisib plus fulvestrant for patients treated with CDK4/6 inhibitors [83]. Other studies have attempted to use alpelisib for neoadjuvant treatment of breast cancer, but the results of the NEO-ORB phase II study showed that alpelisib combined with letrozole for ER+/HER2− and early-stage breast cancer had no additional clinical benefit [84]. Taselisib inhibits p110α, δ and γ, but is 30-fold less potent against p110β. The SANDPIPER phase III clinical trial evaluated the safety of taselisib plus fulvestrant for postmenopausal breast cancer with disease recurrence/progression during or after an aromatase inhibitor. This trial indicated that taselisib increased the frequency of grade 3–5 adverse events (16.4% in placebo arm vs. 49.5% in taselisib arm). Based on this trail, taselisib combined with fulvestrant was discontinued, suggesting that follow-up PI3K inhibitors should improve selectivity for key isoforms, not just selectivity [85]. Other compounds, such as GDC0077 [93], eganelisib and samotolisib are also in preclinical studies.

Dual PI3K-mTOR inhibitors can target the catalytic pockets of mTOR and PI3K enzymes based on structural similarity. Since mTOR inhibitors may enhance the PI3K/PDK1 axis, an inhibitor targeting both PI3K and mTOR may have better anti-cancer activity [94]. Gedatolisib (PF-05212384) is the represent drug of dual PI3K-mTOR inhibitors. Preclinical studies have demonstrated that gedatolisib combined with letrozole/palbociclib or fulvestrant/palbociclib has antitumor activity with manageable toxicity [86,95]. In addition, a phase I trial of samotolisib (dual PI3K-mTOR inhibitors) in combination with cyclin dependent kinase (CDK) inhibitors in ER+ breast cancer patients is ongoing [88]. Recent studies indicate that PI3K/mTOR inhibitors combined with paclitaxel can enhance tumor response to immunosuppressants and may provide a viable treatment for metastatic breast cancer. Therefore, dual target inhibitors in combination with immunotherapy will also be the focus of future research [87,88,96].

#### 3.1.2. AKT Inhibitors for Treating ER+/HER2− Breast Cancer

There are different kinds of AKT inhibitors. Pan-AKT inhibitors bind to the ATP pocket of AKT1/2/3 and suppress their activity [97] (Table 3). Another kind of AKT inhibitor is allosteric inhibitor [98]. Allosteric Akt inhibitor such as MK-2206 is a kind of PH-domain dependent inhibitor [99]. A phase II trial (NCT01776008) showed that MK-2206 did not increase the efficacy of anastrozole monotherapy in patients with PIK3CA-mutated ER+ breast cancer [100].

Capivasertib (AZD5363) is another inhibitor of all three AKT isoforms [101]. A randomized study assessed the effects of capivasertib plus fulvestrant in ER+, HER2− advanced breast cancer patients resistant to endocrine therapy (FAKTION). This trial and its updated analysis showed that the addition of capivasertib to fulvestrant resulted in a significant improvement of progression-free survival, objective response rate (ORR) [102] and overall survival [103] in participants with aromatase inhibitor-resistant ER-positive, HER2−negative advanced breast cancer. Additionally, the expanded biomarker testing suggested that capivasertib was predominantly effective in patients with PI3K/AKT/PTEN pathway-altered tumors (38.9 vs. 20.0 months, *p* = 0.0047) [103]. The grade 3–4 adverse events were hypertension (capivasertib arm vs. placebo: 32% vs. 24%), diarrhea (14% vs. 4%) and rash (20% vs. 0) [102]. Further study and stricter monitoring and management of adverse reactions are needed. Moreover, a basket trial of capivasertib treatment of patients with AKT1 (E17K)-mutated tumors demonstrated an objective response rate of 33%, with clinical benefit in ER+/HER2− breast cancer [111]. Multiple preclinical studies have confirmed that the combination of PARP and PI3K/AKT pathway inhibitors has synergistic antitumor activity in breast cancer susceptibility gene (BRCA)-deficient cancer models [112]. Phase I trials of olaparib, a PARP inhibitor, and capivasertib in BRCA1/2 and non-BRCA1/2 mutated breast cancer patients are ongoing [113].

#### 3.1.3. mTOR Inhibitors for Treating ER+/HER2− Breast Cancer

It is known that mTOR can phosphorylate ERα at ser118, making it insensitive to tamoxifen [71,114]. The BOLERO-2 study showed that the median progression-free survival in postmenopausal ER+/HER2− breast cancer patients resistant to aromatase inhibitor was improved by everolimus in combination with exemestane compared to placebo and exemestane [104]. These findings prompted the FDA to approve everolimus for the patients with ER+, HER2− advanced disease or its combination with exemestane for the treatment of relapse or progression after the use of nonsteroidal aromatase inhibitors in postmenopausal ER+/HER2− advanced breast cancer patients without visceral disease [25,105,106,108]. The European Medicines Agency (EMA) also approved everolimus for ER+/HER2− advanced breast cancer patients after failure of non-steroidal aromatase inhibitors treatment. Exploration of everolimus in the neoadjuvant offsetting for breast cancer has yielded initial results. A study of everolimus plus letrozole for preoperative neoadjuvant treatment of breast cancer was conducted. Everolimus combined with letrozole resulted in a superior response and inhibition of tumor proliferation than letrozole alone [110,115]. However, further studies are required to confirm the effectiveness and safety of the drug.

Sapanisertib (MLN0128) has dual specificity for the mTOR complex (mTORC1 and mTORC2), and is a new generation of ATP-competitive mTOR kinase inhibitors. A phase II study of sapanisertib in combination with exemestane or fulvestrant in postmenopausal women with previously treated everolimus-sensitive or-resistant breast cancer was conducted. It was well tolerated and showed significant clinical benefit [109]. However, follow-up research is necessary. Some studies have found that compensatory IGF signaling could reduce the effectiveness of mTOR inhibitor in combination with endocrine therapy [108]. Further studies may focus on the effects of combining IGF axis inhibitors and mTOR inhibitors.

### 3.2. PAM Inhibitors for Treating HER2+ Breast Cancer

HER2 is overexpressed in about 20–25% of breast cancers [52]. HER2−targeted therapy is the standard of care for these patients [26]. PAM pathway is one of the major signaling pathway downstream of HER2. Its abnormal activation, such as PIK3CA mutation and constitutively active AKT, is involved in the development of primary and secondary resistance to HER2− targeted therapy [116,117]. PAM pathway inhibitors may help restore tumor sensitivity to anti-HER2 therapy (Table 4), but the efficacy and safety need to be further explored.

Buparlisib is a pan-PI3K inhibitor [118,119]. A phase II trial of buparlisib plus trastuzumab in trastuzumab-resistant, HER2−positive advanced breast cancer patients was conducted. Similar to many pan-PI3K inhibitors, the common adverse reactions of buparlisib were diarrhea (54%) and nausea (48%). Unfortunately, this trial failed to demonstrate a benefit of addling buparlisib to trastuzumab [119]. Copanlisib (BAY80-6946), a PI3Kα/δ inhibitor, was shown to be synergistic with anti-HER2 therapy in trastuzumab-resistant breast cancer cells [125]. A phase Ib trial indicated that copanlisib in combination with trastuzumab was well-tolerated in HER2+ metastatic breast cancer patients [126]. In a phase I trial (BYL-719) for alpelisib in combination with T-DM1(Ado-trastuzumab emtansine) in trastuzumab-resistant and/or T-DM1-resistant, HER2+ and metastatic breast cancer patients, the CBR (CR + PR) in the entire patient population and patients with prior T-DM1 treatment was 71% and 60%, respectively [120]. To further improve the efficacy and safety, specific PI3K inhibitors are still in development. Of note, GDC0941 and XL-147(pan-PI3K inhibitors) or BEZ235 (dual PI3K/mTOR inhibitor) can increase HER2/3 expression thereby promoting RAS/MAPK (mitogen-activated protein kinases) signaling [127,128,129]. Hence, PAM inhibitors combined with MEK inhibitors may be useful to overcome drug resistance [130].

The combination of MK-2206 with standard neoadjuvant therapy given rise to pCR rates in breast cancer patients with HER2 over expression [121]. However, MK-2206 has not yet been developed further. AKT/mTOR is highly activated in lapatinib-resistant HER2+ breast cancer cells [131]. The mTOR inhibitor INK-128 can restore the sensitivity of lapatinib-resistant HER2+ breast cancer cells to TKIs [127]. Recent study indicates that everolimus combined with T-DM1 has a strong in vivo and in vitro antitumor effect on HER2 positive breast cancer [132]. In a phase II clinical trial, the mTOR inhibitor sirolimus combined with trastuzumab was well tolerated in patients with trastuzumab-resistant, HER2-positive and advanced breast cancer [124]. The phase III BOLERO-3 study of everolimus combined with vinorelbine and trastuzumab in HER2-positive, trastuzumab-resistant advanced breast cancer was conducted. Compared with the placebo arm, everolimus combined with vinorelbine and trastuzumab significantly prolonged the mPFS (5.78 vs. 7.0 month; *p* = 0.0067) [123]. The exploratory analysis found that patients with PIK3CA mutations, PTEN deletions or tumors with an overactive PI3K pathway could gain PFS benefit from everolimus [122]. However, further studies are still in progress [124].

### 3.3. PAM Inhibitors for Treating Triple-Negative Breast Cancer (TNBC)

TNBC has a high degree of malignancy and rapid metastasis [52]. Abnormal activation of the PAM pathway is also common in TNBC, especially the type of luminal androgen receptor (LAR) in the Fudan classification [133]. This population may benefit from PI3K-targeted therapy (Table 5). A trial of the pan-PI3K inhibitor buparlisib in metastatic TNBC patients was conducted. Although the downregulation of vital components in the PI3K pathway was observed in patients with stable disease, the trial failed to observe a clear objective response. Inhibition of PI3K alone may not be sufficient for treating TNBC [134]. A phase II study of the sequential treatment of metastatic TNBC with PI3K-α inhibitor serabelisib, cisplatin and nab-paclitaxel is underway. In addition, a phase I study of eganelisib (dual PI3K-mTOR inhibitor) for treating advanced or metastatic TNBC is under recruitment.

AKT inhibitors combined with paclitaxel have shown remarkable antitumor activity as first-line drugs in the treatment of metastatic breast cancer. In the phase II PAKT study of the AKT1-3 isoform inhibitor capivasertib combined with paclitaxel as a first-line treatment of TNBC, combination of capivasertib and paclitaxel significantly prolonged PFS and OS compared with paclitaxel alone (*p* = 0.04, Table 5). The difference in survival benefit was more obvious in patients with PIK3CA/AKT1/PTEN-altered tumors (9.3 vs. 3.7 months; *p* = 0.01) [135]. In the LOTU phase II study for first-line treatment of TNBC, the pan-AKT inhibitor GDC-0068 (Ipatasertib) also showed an advantage in prolonging PFS compared with paclitaxel alone (*p* = 0.037) [136]. Another ongoing phase II trial, FAIRLANE, is also exploring the efficacy and safety of GDC-0068 in combination with paclitaxel in patients with grade IA-IIIA TNBC [137]. Some studies have confirmed that suppressing AKT and/or p70S6K (p70 ribosomal protein S6 kinase) activation might synergize with paclitaxel [141]. LY2780301 is a dual inhibitor of p70S6K and AKT. The phase Ib/II TAKTIC trial aims to evaluate LY2780301 in combination with weekly paclitaxel for treating HER2-negative advanced breast cancer patients. The combination of LY2780301 and paclitaxel demonstrated ORR benefit, while the main grade 3–4 drug-related adverse events included neuropathy (8%) and ungual toxicity (25%) [138].

The mTOR inhibitor everolimus was also used in the neoadjuvant treatment of TNBC. However, compared with the placebo group, combination of everolimus with cisplatin and paclitaxel did not demonstrate CR benefit (pCR: 36% vs. 49%) [139]. A study of tamirolimus or everolimus in combination with doxorubicin and bevacizumab for the treatment of metastatic TNBC showed initial results. mTOR inhibitors prolonged the ORR to 21%, and improved the 6-month clinical benefit rate to 40% [140]. The objective response benefit was associated with PI3K pathway aberration (*p* = 0.04) [140]. The status of PTEN may be a biomarker of PAM inhibitors for TNBC [142].

## 4. Conclusions and Perspectives

The PI3K/AKT/mTOR pathway adjusts cell proliferation and metabolism. Abnormalities in key targets lead to over-activation of this pathway, which is involved in tumor development, progression and drug resistance. PAM pathway inhibitors may be reliable antitumor agents (Table 6). At present, the PI3K inhibitor alpelisib and mTOR inhibitor everolimus have been approved by the FDA and EMA for the treatment of advanced ER+ breast cancer. The new generation of AKT inhibitors capivasertib and the highly selective ATP-competitive mTOR kinase inhibitors sapanisertib and sirolimus remain to be extensively evaluated. Most PAM inhibitors have limited effectiveness due to weak isoform selection inhibition, feedback regulation of downstream pathways, and tandem interference with other signaling pathways. Perhaps stronger PI3K subtype-specific inhibitors such as GDC-0077(p110α inhibitor) may show a better antitumor efficacy. In addition, the novel AKT inhibitor INY-03-041, which is composed of Ipatasertib-NH2, a ten-hydrocarbon linker and a cereblon ligand lenalidomide, can target all three AKT protein for proteasomal degradation [143]. More and more AKT degraders have been developed [144]. Preclinical studies demonstrate that these AKT degraders can effectively suppress tumor growth [144]. It remains to know whether these compounds are tolerable and have superior efficacy in clinical setting.

In conclusion, different combination treatments may be considered for different breast cancer subtypes. In ER+ breast cancer patients, PAM inhibitors combined with CDK inhibitors or other anti-estrogen therapies are expected to overcome drug resistance and prolong survival. For breast cancer with HER2-overexpressed breast cancer, the combination of mTOR inhibitor with anti-HER2 drugs (trastuzumab, TDM-1) may be an alternative after the progression of second-line therapy. For TNBC, concurrent targeting PAM and other key pathway nodes (EGFR, MEK) may be of benefit. Moreover, it remains to evaluate the efficacy of combined treatment of breast cancer patients with immune checkpoint inhibitors and PAM inhibitors. Currently, the combination of PARP and AKT inhibitors has also shown prospective results in breast cancer patients with BRCA1/2 mutation [145,146]. Future studies should focus on improving subtype selectivity and reducing toxic side effects. The combination of different PAM pathway inhibitors, or a combination with immunotherapy drugs, may also be a strategy to further improve efficacy [147]. Screening for reliable biomarkers that predict the effectiveness of combination regiments is also critical.

## Figures and Tables

**Figure 1 cells-11-02508-f001:**
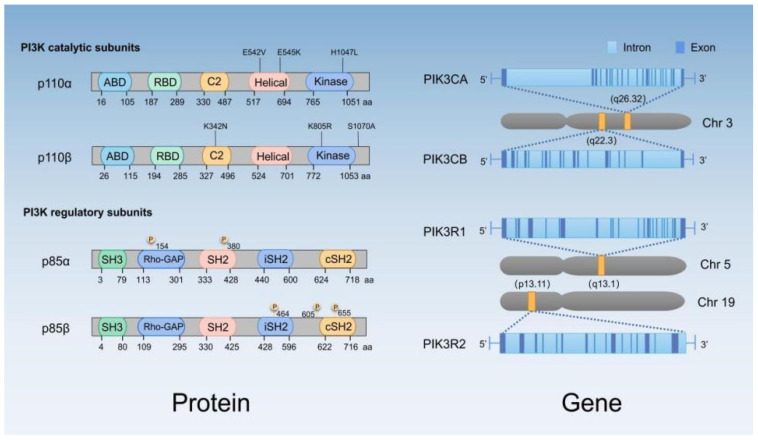
The genes and proteins of class I PI3K. The catalytic subunits p110α/p110β, the regulatory subunits p85α/p85β and the genes that encoded these subunits (PIK3CA, PIK3CB, PIK3R1 and PIK3R2) are shown. ABD, albumin binding domain; RBD, RAS-binding domain; C2, protein kinase C conserved region 2; Rho-GAP, Rho GTPase-activating protein; SH2, Src homology 2 domain; iSH2, inter-SH2; cSH2, C-terminal SH2.

**Figure 2 cells-11-02508-f002:**
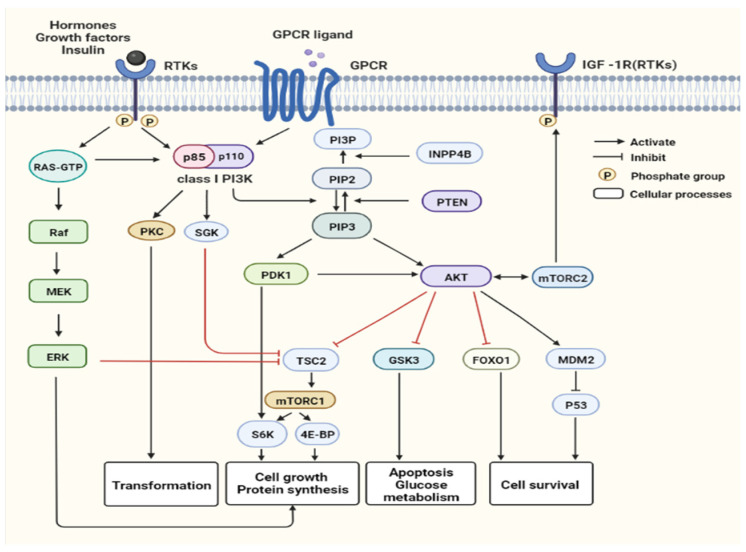
The PAM signaling pathways. PAM pathway can be activated by G protein-coupled receptor (GPCR) and receptor tyrosine kinases (RTKs), including human epidermal growth factor receptor 2 (HER2/ERBB2), fibroblast growth factor receptor (FGFR), insulin and insulin-like growth factor-1 receptor (InsR/IGF-1R), which PtdIns (4,5) P2 (PIP2) to generate the second messenger PtdIns (3,4,5) P3 (PIP3) [6]. PTEN (phosphatase and tensin homolog) dephosphorylates PIP3 to generate PIP2, while INPP4B (inositol polyphosphate-4-phosphatase type II B) dephosphorylates PtdIns(3,4)P2 to generate PtdIns(3)P (PI3P). Proteins, containing a pleckstrin homology (PH) domain, are recruited to the cytomembrane by PIP3, including AKT, 3-phosphoinositide-dependent kinase 1 (PDK1), and serum and glucocorticoid-induced kinase (SGK). The main downstream target of PI3K is AKT. It is activated by PDK1 and mTOR complex 2 (mTORC2), and phosphorylates a large number of downstream effector proteins, including mTOR complex 1 (mTORC1), forkhead box protein O1 (FoxO1), glycogen synthesis kinase (GSK3) and murine double minute 2 (MDM2). The AKT-mediated phosphorylation of GSK3β, FOXO1 and MDM2 directly or indirectly controls cellular growth and survival. The activated mTORC1 ultimately regulates cellular processes, such as the initiation of mRNA transcription, cell growth, autophagy and protein synthesis, via phosphorylation of 4EBP1 and S6K1 [1,7]. In addition, mTORC2 can phosphorylate both IGF-1R and AKT [8]. S6K-mediated phosphorylation of PDK1 negatively feed back to inhibit PDK1 [7]. PKC and SGK are also involved in PI3K signaling independent of AKT.

**Table 1 cells-11-02508-t001:** Frequencies of changes of PAM pathway in different molecular subtypes of breast cancer.

Gene (Protein)	Alteration	Effect on Signaling	Correlation with Prognosis	Frequency				Reference
				Luminal (ER+)		HER2+	TNBC (ER−, PR−, HER2−)	
				A	B			
PTEN	Inactivation and mutation/reduced expression	over activation of PI3K signaling	Negative in TNBC	29–44%		22%	67%	[33,34,35]
PIK3CA (p110α/PI3K)	Activating mutation	Hyperactivation of PI3K signaling	* Positive in luminal, negative in metastatic/HER2+ breast cancer	47%	33%	23–39%	8–25%	[33,36,37,38]
PIK3CB (p110β/PI3K)	Amplification /Mutation	PIP3 accumulates and activates AKT	Irrelevant	5%				[29,39]
PIK3R1 (p85α/PI3K)	Inactivating mutation	Derepression of catalytic activity of p110α	-	2% of Early breast cancer 11% of Metastatic breast cancer				[40,41]
AKT1	Activating mutation	Hyperactivation of AKT	Irrelevant	2.6–7.4%				[33,42,43,44]
AKT2	Amplification		Irrelevant	2.8–4%				[18]
AKT3	Amplification		Positive in luminal A breast cancer	15%				[40]
PDK1	Amplification	Hyperactivation of AKT	-	20–38%				[45]
(mTOR)	p-mTOR expression	Hyperactivation of mTOR	Negative in TNBC	39%			37.5–72.1%	[46]

* Positive: Associated with a better prognosis; Negative: Associated with a worse prognosis; Irrelevant: No significant correlation with prognosis; p-mTOR: phosphorylated mTOR.

**Table 2 cells-11-02508-t002:** Clinical trials of PI3K inhibitors in ER+/HER2− breast cancer.

Target	Drug	Study (Phase)	Patient Population	Regimen and Outcome	FDA/EMA Approval	Reference
Pan-PI3K	Buparisib (BKMI20)	BELLE-2 (III)	HR (+), HER2 (−), ABC/MBC (second line)	buparlisib + fulvestrant vs. placebo + fulvestrant (mPFS: 6.9 vs. 5.0 months; HR: 0.78; *p* = 0.00021)	N	[78]
		BELLE-3 (II)	HR (+), HER2 (−), ABC/MBC relapsed on or after endocrine therapy and mTOR inhibitors	buparlisib vs. Placebo (mPFS: 3.9 vs. 1.8 months; HR: 0.67; *p* = 0.0003)		[79]
	Pictilisib (GDC-0941)	FERGI (II)	HR (+), HER2 (−), ABC/MBC Al-resistant	pictilisib + fulvestrant vs. placebo + fulvestrant (mPFS:6.6 vs. 5.1 months; HR: 0.74; *p* = 0.096)	N	[80]
		PEGGY (II)	HR (+), HER2 (−) metastatic breast cancer	Pictilisib + paclitaxel vs. placebo + paclitaxel (mPFS:8.2 vs. 7.8 months; HR: 0.95)		[81]
PI3K (p110α)	Alpelisib (BYL719)	SOLAR-1 (III)	HR (+), HER2 (−), ABC Received endocrine therapy previously	PIK3CA-mutated: alpelisib vs. placebo (mPFS 11.0 vs. 5.7 months; HR: 0.65; *p* < 0.001); (mOS: 39.3 vs. 31.4 months; HR: 0.86; *p* = 0.15)	Y	[82]
		BYLieve (II)	HR (+), HER2 (−), PIK3CA-mutant ABC progressed on/after prior therapy, including CDK inhibitors	proportion of without disease progression at 6 month was 50.4% (95% CI: 41.2–59.6).		[83]
		NEO-ORB (II)	HR (+), HER2 (−) Postmenopausal women Tlc-T3 breast cancer	Alpelisib + letrozole vs. placebo + letrozde, ORR: 43% vs. 45%, PIK3CA-wild-type vs. mutant ORR: 63% vs. 61%		[84]
	Taselisib (GDC0032)	SANDPIPER (III)	Postmenopausal women, disease recurrence/progression during/after AI	Taselisib vs. placebo (PFS: 7.4 vs. 5.4 months; HR: 0.70: *p* = 0.0037)	N	[85]
PI3K-mTOR	Gedatolisib	NCT02684032 (I)	metastatic breast cancer	NA	N	[86]
	Apitolisib	NCT01254526 (Ib)	locally recurrent breast cancer or metastatic breast cancer	NA	N	[87]
	Samotolisib	NCT02057133 (I)	In combination with: letrozole, anastrozole, tamoxifen, exemestane	NA	N	[88]

Al, aromatase inhibitor; mPFS, median progression-free survival; HR, hazard ratio: MBC, metastatic breast cancer. ABC, advanced breast cancer; ORR, objective response rate; NA, not applicable or discontinued owing to drug toxicity; EMA, European Medicines Agency; N, not yet approved; Y, approved. FDA, Food and Drug Administration.

**Table 3 cells-11-02508-t003:** Clinical trials of AKT and mTOR inhibitors in ER+/HER2− breast cancer.

Target	Drug	Study (Phase)	Patient Population	Regimen and Outcome	FDA/EMA Approved	Reference
AKT1-3	Capivasertib (AZD5363)	BEECH (II)	ER (+), HER2 (−) ABC/MBC (first-line)	capivasertib + paclitaxel vs. placebo + paclitaxel (mPFS: 10.9 vs. 8.4 months; HR: 0.80; *p* = 0.308) PIK3CA+ sub-population (mPFS: 10.9 vs. 10.8 months; HR: 1.11; *p* = 0.760)	N	[101]
		FAKTION (II)	ER (+)/HER (−) ABC/MBC; Postmenopausal relapsed or progressed on AI	capivasertib + fulvestrant vs. placebo + fulvestrant (mPFS: 10.3 vs. 4.8 months; HR: 0.58; *p* = 0.0044; mOS: 29.3 vs. 23.4 months; HR: 0.66; *p* = 0.035)		[102,103]
Pan-AKT	MK-2206	NCT01776008 (II)	Endocrine resistant, ER+ breast cancer	0% pCR	N	-
mTOR (mTORC1)	Everolimus (RAD001)	BOLERO-2 (III)	ER (+)/HER2−, AI-resistant and postmenopausal ABC	Everolimus + exemestrane vs. placebo + exemestrane (final PFS:11.0 vs. 4.1 months; HR: 0.38; *p* < 0.0001)		[104]
		MANTA (II)	HR+, postmenopausal and AI-resistant locally ABC or MBC	Everolimus +fulvestrant vs. fulvestrant (mPFS: 12.3 vs. 5.4 months; HR: 0.63; *p* = 0.01)	A	[105]
		PrE0102 (II)	ER (+)/HER2−, AI-resistant and postmenopausal MBC	Everolimus + fulvestrant vs. placebo + fulvestrant (mPFS:10.3 vs.5.1 months; HR: 0.61; *p* = 0.02) ORR: 18.2 vs. 12.3%; *p* = 0.47		[106]
		BOLERO-6 (III)	ER (+)/HER2−ABC	Everolimus + exemestrane vs. Exemestrane vs. capecitabine mOS: 23.1 vs. 29.3 months vs. 25.6 months		[107]
		NCT02123823 (I-II)	ER (+)/HER2− ABC and MBC	xentuzumab + everolimus + exemestane, vs. exemestane + everolimus (mPFS: 7.3 vs. 5.6 months; *p* = 0.9057)		[108]
mTOR (mTORC1/2)	MLN0128	NCT02049957 (I-II)	HR+/HER2− and AI-resistant MBC	everolimus-sensitive vs. everolimus-resistant cohorts, ^1^ CBR-16: 45% vs. 23%, ^2^ ORR: 8% vs. 2%	N	[109]
	Apanisertib	NCT02756364 (II)	HR+/HER2− and AI-resistant MBC	NA	N	-
Pan-mTOR	Temsirolimus	HORIZON (III)	HR+, postmenopausal and AI-naïve ABC	Temsirolimus vs. placebo + letrozole (mPFS: 9.0 vs. 5.6 months; HR: 0.7, *p* < 0.009)	N	[110]

^1^ CBR-16, clinical benefit rate at 16 weeks; ^2^ ORR, overall response rate.

**Table 4 cells-11-02508-t004:** Clinical trials of PAM inhibitors in HER2+ breast cancer.

Target	Drug	Study (Phase)	Patient Population	Regimen and Outcome	Reference
Pan-PI3K	Buparlisib (BKM120)	BKM120 (II)	Trastuzumab-resistant HER2+ breast cancer	buparlisib + trastuzumab: ORR:10% (ORR ≥ 25%)	[118]
		PIKHER (II)	Trastuzumab-resistant HER2+ ABC	DCR: 79%; 95% CI: 57–92%, CBR: 29%; 95% CI: 12–51%.	[119]
	Alpelisib	BYL-719 (I)	Trastuzumab- and taxane-resistant HER2+ MBC	Alpelisib + T-DM1: ORR: 43%. T-DM1-resistant (n = 10): ORR 30%. mPFS 8.1 months	[120]
Pan-Akt	MK-2206	SPY2 (II)	High-risk, early-stage Breast cancer with neoadjuvant therapy	MK-2206 vs. control: pCR 61.8% vs. 35% (control: standard taxane- and anthracycline-based neoadjuvant therapy)	[121]
AKT-1	Ipatasertib (IPAT)	SOLTI-1507 (Ib)	HER2+ ABC or MBC with PIK3CA mutation	NA	-
mTOR	Everolimus	BOLERO-1 (III)	HER2+, HR- primary ABC	Everolimus vs. placebo + trastuzumab mPFS: 20.3 vs.13.1 months; HR: 0.66; *p* = 0.0049	[122]
		BOLERO-3 (III)	Taxane-pretreated and trastuzumab-resistant HER2+ ABC	Everolimus vs. Placebo + trastuzumab, vinorelbine mPFS: 7.0 vs.5.8 months; HR: 0.78; *p* = 0.0067	[123]
	Sirolimus	M124188 (II)	HER2(+) MBC	Trastuzumab + sirolimus ORR: 1/9 (11%) CBR: 4/9 (44%)	[124]

**Table 5 cells-11-02508-t005:** Clinical trials of PAM inhibitors in triple negative breast cancer (TNBC).

Target	Drug	Study (Phase)	Patient Population	Regimen and Outcome	Reference
Pan-PI3K	Buparlisib (BKM120)	NCT01790932 (II)	Metastatic TNBC	CBR:12% (6 patients, all SD ≥ 4 months) mPFS: 1.8 months (95% CI: 1.6–2.3) mOS: 11.2 months (95% CI: 6.2–25)	[134]
AKT1-3	Capivasertib	(II)	Metastatic TNBC	Capivasertib vs. Placebo + paclitaxel (mPFS: 5.9 vs. 4.2 months; *p* = 0.06) (mOS 19.1 vs. 12.6 months; *p* = 0.04)	[135]
Pan-AKT	GDC-0068 (Ipatasertib)	LOTU (II)	Metastatic TNBC	Ipatasertib vs. placebo + paclitaxel (mPFS 6.2 vs. 4.9 months; HR: 0.60; *p* = 0.037)	[136]
		FAIRLANE (II)	Early TNBC	Ipatasertib + paclitaxel vs. placebo + paclitaxel pCR rates: 17% vs. 13%	[137]
	LY2780301	TAKTIC (Ib/II)	HER2-ABC	6-month ORR:63.9% [48.8–76.8]	[138]
mTOR	Everolimus (DAE)	NCT00930930 (II)	II/III TNBC (Neoadjuvant therapy)	Everolimus vs. placebo (^5^pCR: 36% vs. 49%)	[139]
	Temsirolimus (DAT)	NCT00761644 (II)	Metaplastic TNBC	Doxorubicin + bevacizumab + DAT or DAE ORR: 21%; CBR: 40% PI3K pathway alteration: ORR: (31% vs. 0%; *p* = 0.04); CBR (44% vs. 45%; *p* > 0.99).	[140]
PI3K-mTOR	Eganelisib	NCT03719326 (I/Ib)	Advanced or metastatic TNBC	In combination with pegylated liposomal doxorubicin (PLD) or A2aR/A2bR antagonist-1(AB928) NA	-

CI, confident interval; SD, stable disease; mOS, median overall survival; mPFS, median Progression-Free Survival; pCR, pathological complete response; NA: not available.

**Table 6 cells-11-02508-t006:** Lists of clinical trials of some PAM inhibitors in different subtypes of breast cancer.

Target	Drug	HR+, HER2−	HER2+	Triple Negative Breast Cancer
Class I PI3K	Buparisib	NCT01633060	-	NCT01790932
		NCT01339442		
		NCT01610284		
	Pictilisib	NCT01437566	NCT00928330	NCT01918306
		NCT01740336		
	Alpelisib	NCT02379247	NCT05230810	NCT02038010
		NCT03386162		
		NCT04208178		
	Taselisib	NCT02340221	NCT02390427	NCT02457910
		NCT02273973		
PI3K-mTOR	Gedatolisib	NCT02684032	NCT03698383	NCT03243331
				NCT01920061
	Apitolisib	NCT01254526	-	-
	Samotolisib	NCT02057133	-	-
	Eganelisib	-	-	NCT03961698
AKT	Capivasertib	NCT01277757	-	NCT03997123
		NCT01992952		NCT03742102
	MK-2206	NCT01776008	-	-
	Ipatasertib	-	NCT03840200	NCT02301988
				NCT03800836
	LY2780301	-	-	NCT01980277
mTOR	Everolimus	NCT02216786	NCT00912340	NCT01931163
		NCT01797120	NCT00876395	
		NCT01783444		
		NCT02123823		
	MLN0128	NCT02049957	-	NCT02719691
		NCT02988986		
	Apanisertib	NCT02756364	-	-
	Temsirolimus	NCT02152943	NCT00411788	NCT02723877
		NCT01248494		
	Sirolimus	NCT00411788	NCT01783444	-
	Ridaforolimus	-	NCT00736970	-

## Data Availability

Not applicable.

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
