# Peer review of "PI3K/AKT/mTOR-Targeted Therapy for Breast Cancer"

_cells, 2022, doi:10.3390/cells11162508_

Round 1

Reviewer 1 Report

The manuscript is concisely written, and the research question is original. The methods were appropriate for the research question asked and the experiments were properly controlled. In my opinion, this study addresses a significant problem, and the aim and the application are achieved. I think that this report is acceptance.

Author Response

Thanks for the recognition of our manuscript.

Reviewer 2 Report

This manuscript is well written and in good organization. The authors may add some updated refs e.g. 'Updated FAKTION data showed that capivasertib addition to fulvestrant extends the survival of participants with aromatase inhibitor-resistant ER-positive, HER2-negative advanced breast cancer. The expanded biomarker testing suggested that capivasertib predominantly benefits patients with PI3K/AKT/PTEN pathway-altered tumours.' Lancet Oncol. 2022 Jul;23(7):851-864.; 'Candidates for a regimen with a phosphatidylinositol 3-kinase inhibitor and hormonal therapy should undergo testing for PIK3CA mutations using next-generation sequencing of tumor tissue or circulating tumor DNA (ctDNA) in plasma to determine eligibility for alpelisib plus fulvestrant. J Clin Oncol Jun 27 2022

2022 Jun 27

tumor DNA (ctDNA) in plasma to determine eligibility for alpelisib plus fulvestrant.

Author Response

Thank you for evaluating our manuscript. The response to your concern is as following:

Point: The authors may add some updated refs e.g. 'Updated FAKTION data showed that capivasertib addition to fulvestrant extends the survival of participants with aromatase inhibitor-resistant ER-positive, HER2-negative advanced breast cancer. The expanded biomarker testing suggested that capivasertib predominantly benefits patients with PI3K/AKT/PTEN pathway-altered tumours.' Lancet Oncol. 2022 Jul;23(7):851-864.; 'Candidates for a regimen with a phosphatidylinositol 3-kinase inhibitor and hormonal therapy should undergo testing for PIK3CA mutations using next-generation sequencing of tumor tissue or circulating tumor DNA (ctDNA) in plasma to determine eligibility for alpelisib plus fulvestrant. J Clin Oncol Jun 27 2022.8.

 Response: Here are the two updates we've added:

Page 7, lines 215-220: The recently updated American Society of Clinical Oncology (ASCO) guideline recommended that patients with locally recurrent unresectable or metastatic hormone receptor positive and HER2-negative breast cancer should undergo testing for PIK3CA mutations to determine their eligibility for treatment with the alpelisib plus fulvestrant

Page 9, lines 259-265: “This trial and its updated analysis showed that the addition of capivasertib to fulvestrant resulted in a significant improve of progression-free survival, objective response rate (ORR) and overall survival in participants with aromatase inhibitor-resistant ER-positive, HER2-negative advanced breast cancer. Additionally, the expanded biomarker testing suggested that capivasertib predominantly benefits patients with PI3K/AKT/PTEN pathway-altered tumors (38.9 vs. 20.0 months, p=0.0047)”.

Reviewer 3 Report

In the manuscript “PI3K/AKT/mTOR-targeted therapy for breast cancer”, the authors collect and organize the data about the role and regulation of PI3K/AKT/mTOR pathway in breast tumors, summarizing, in particular, the changes in this intracellular signaling discovered in different subtypes of breast cancer.

The review is clear in most of its parts, is well organized with an updated bibliography and the language is precise.

As a unique point, I would suggest the addition in the conclusion of a personal interpretation of the relevance of this pathways in breast cancer highlighting, for example, that the absence of AKT inhibitors specifically directed against the 3 isoform could be a great obstacle in the efficacy of this targeted pathway.

Author Response

Thank the reviewer for evaluating our manuscript. The response to your concern is as following:

Point: As a unique point, I would suggest the addition in the conclusion of a personal interpretation of the relevance of this pathways in breast cancer highlighting, for example, that the absence of AKT inhibitors specifically directed against the 3 isoform could be a great obstacle in the efficacy of this targeted pathway.

Response: As the reviewer suggested, we edited the section of Conclusion. Recently, some AKT degraers have been developed. The PROTAC AKT degrader INY-03-041 reportedly targets all three isoforms for degradation. We introduced this point in the revised manuscript. The following statement was added into the revised manuscript: “Perhaps stronger PI3K subtype-specific inhibitors such as GDC-0077(p110α inhibitor) may show a better antitumor efficacy. In addition, the novel AKT inhibitor INY-03-041, which is composed of Ipatasertib-NH2, a ten-hydrocarbon linker, and a cereblon ligand lenalidomide, can target all three AKT protein for proteasomal degradation. More and more AKT degraders have been developed. Preclinical studies demonstrate that these AKT degraders can effectively suppress tumor growth. It remains to know whether these compounds are tolerable and have superior efficacy in clinical setting”.

Reviewer 4 Report

In this review article, the authors have described the changes of PI3K/AKT/mTOR pathway in different breast cancer subtypes,  and summarized current advance of targeting this pathway in both preclinical and clinical studies. This article is well written and has provided an overview of PI3K/AKT/mTOR targeted therapy in breast cancer.

Author Response

Thank you for your comments and recognition of our manuscript.